# Nanolipid-Trehalose Conjugates and Nano-Assemblies as Putative Autophagy Inducers

**DOI:** 10.3390/pharmaceutics11080422

**Published:** 2019-08-20

**Authors:** Eleonora Colombo, Michele Biocotino, Giulia Frapporti, Pietro Randazzo, Michael S. Christodoulou, Giovanni Piccoli, Laura Polito, Pierfausto Seneci, Daniele Passarella

**Affiliations:** 1Dipartimento di Chimica, Università degli Studi di Milano, Via Golgi 19, 20133 Milano, Italy; 2CIBIO, Università di Trento, Via Sommarive 9, 38123 Povo (TN), Italy; 3Promidis Srl, San Raffaele Scientific Research Park, Torre San Michele 1, Via Olgettina 60, 20132 Milan, Italy; 4DISFARM, Sezione di Chimica Generale e Organica “A. Marchesini”, Universitdegli Studi di Milano, via Venezian 21, 20133 Milano, Italy; 5ISTM-CNR, via Fantoli 16/15, 20138 Milan, Italy

**Keywords:** nano-assemblies, trehalose, squalene, betulinic acid, autophagy induction

## Abstract

The disaccharide trehalose is an autophagy inducer, but its pharmacological application is severely limited by its poor pharmacokinetics properties. Thus, trehalose was coupled via suitable spacers with squalene (in 1:2 and 1:1 stoichiometry) and with betulinic acid (1:2 stoichiometry), in order to yield the corresponding nanolipid-trehalose conjugates **1-Sq-mono**, **2-Sq-bis** and **3-Be-mono**. The conjugates were assembled to produce the corresponding nano-assemblies (NAs) **Sq-NA1**, **Sq-NA2** and **Be-NA3**. The synthetic and assembly protocols are described in detail. The resulting NAs were characterized in terms of loading and structure, and tested in vitro for their capability to induce autophagy. Our results are presented and thoroughly commented upon.

## 1. Introduction

Nano-vectors are used as therapeutics and diagnostics [1,2]. Iron-based [3], gold-based [4] and silica-based inorganic nano-vectors [5] were tested as being a diagnostic (magnetic resonance imaging reagents/MRIs [6]), or as therapeutics (hyperthermia against tumors [7], iron replacement therapies [8]). Liposome- [9], micelle- [10] and polymer-based nano-assemblies (NAs) [11] are marketed, mostly as anti-cancer agents [12]. Exploratory efforts [13] up to clinical trials [14] against diseases of the central nervous system (CNS) were reported.

Trehalose [15,16] is a non-reducing disaccharide made by a 1,1 linkage between two d-glucose molecules. It is bio-synthesized in lower organisms [16] to stabilize life processes and support survival in extreme conditions (freezing [17], heat and desiccation [18]). Trehalose induces autophagy in vitro and in vivo [19] and reduces protein misfolding and aggregation in vitro [20] by acting as a chemical chaperone and solvating them [20]. Reduction of aggregated huntingtin (Huntington Disease) [21], synuclein (Parkinson Disease) [20] and amyloid species (Alzheimer Disease) [22] was observed in vitro. Trehalose was tested as a safe, cheap, neuroprotective agent in preclinical and clinical studies [16]. Unfortunately, high mM trehalose concentrations are needed in vivo for efficacy, due to its high hydrophilicity and due to trehalase enzymes [23], that hydrolyze trehalose in the brush border cells of the small intestine and in the proximal tubules of the kidneys, preventing its oral absorption.

Nano lipid-drug conjugates [24], obtained by the covalent coupling of a drug to bio-compatible lipids, improve pharmacokinetics, decrease toxicity and increase the therapeutic index of the associated drugs. In particular, squalene-based amphiphilic conjugates have a proven track record for therapeutic applications [25]. They spontaneously assemble in water into nano-assemblies (NAs), encasing the bioactive payload, they do not show the drug on their surface, minimizing any side effect [26], they are internalized by cells via endocytic pathways [27] and release the free drug at its site of action, when a biologically labile linkage is used [28]. We also selected betulinic acid-based conjugates, a less studied but promising class of self-assembling NAs [29,30].

In recent years we worked on anticancer drug-containing self-assembling drug conjugates that spontaneously form NAs in aqueous media [31]. We reported NAs composed by conjugate releasable compounds [32]; by single and dual drug fluorescent hetero-NAs [33,34], by dual drug hetero-NAs (cyclopamine/taxol [34], cyclopamine/doxorubicin [35], ecdysteroid/doxorubicin [36]), and by self-assembling conjugate dual drug NAs [37]. We prepared compounds containing a squalene [31,32,33,34,35] or a 4-(1,2-diphenylbut-1-en-1-yl)aniline tail [37,38] that leads to NAs ability to self-assemble in water. We recently reported the assembly and characterization of squalene-thiocolchicine NAs that release cytotoxic, free thiocolchicine in cancer cells through a disulfide bond or a *p*-hydroxybenzyl moiety [38]. Here we describe the synthesis of two squalene-trehalose conjugates **1-Sq-mono** and **2-Sq-bis**, and of a betulinic acid-trehalose conjugate **3-Be-mono**, their assembly and characterization as NAs (respectively **mono-Sq-NA1**, **bis-Sq-NA2** and **mono-Be-NA3**, Figure 1), and their effects in biological assays. We measured cell viability to determine the safety of our NAs, while autophagy induction was selected as a validated neuroprotective mechanism of action in multiple neurodegenerative diseases [39]. Trehalose-containing NAs may significantly increase the weak, high mM effects of trehalose on autophagy induction [19], by facilitating its cellular internalization, and by releasing after NA disassembly/ester hydrolysis.

## 2. Materials and Methods

### 2.1. Synthesis-General

Each reaction was carried out in oven-dried glassware, using dry solvents under a nitrogen atmosphere. Unless otherwise stated, these solvents were purchased from Sigma Aldrich Italy (Milan, Italy) and used without further purification. Chemical reagents were purchased from Sigma Aldrich, and used as such. Thin layer chromatography (TLC) was performed on Merck-pre-coated 60F_254_ plates. Reactions were monitored by TLC on silica gel, with detection by UV light (254 nm), or by charring either with a 1% permanganate solution or a 50% H_2_SO_4_ solution. Flash chromatography columns were run using silica gel (240–400 mesh, Merck Italy, Milan, Italy).

^1^H-NMR spectra were recorded on Bruker DRX-400 and Bruker DRX-300 instruments (Billerica, MA, USA) in either CDCl_3_, CD_3_OD or DMSO-d6. ^13^C-NMR spectra were recorded on the same instrumentation (100 and 75 MHz) in either CDCl_3_, CD_3_OD or DMSO-d6. Chemical shifts (δ) for proton and carbon signals are quoted in parts per million (ppm) relative to tetramethylsilane (TMS), which was used as an internal standard. Electrospray ionization (ESI) MS spectra were recorded with a Waters Micromass Q-Tof micro mass spectrometer (Milford, MA, USA); HR-ESI mass spectra were recorded on a FT-ICR APEX_II_ (Bruker, Billerica, MA, USA), while EI mass spectra were recorded at an ionizing voltage of 6 kEv on a VG 70-70 EQ. Specific rotations were measured with a P-1030-Jasco polarimeter with 10 cm optical path cells and 1 mL capacity (Na lamp, λ = 589 nm).

### 2.2. Synthesis-Squalene-Trehalose Conjugates 1-Sq-Mono and 2-Sq-Bis

1-[(2R,3R,4S,5R,6R)-6-{[(2R,3R,4S,5R,6R)-6-(hydroxymethyl)-3,4,5-tris[(trimethylsilyl)oxy]oxan-2-yl]oxy}-3,4,5-tris[(trimethylsilyl)oxy]oxan-2-yl]methyl10-(3E,7E,11E,15E)-3,7,12,16,20pentamethylheni- cosa-3,7,11,15,19-pentaen-1-yl decanedioate/**15-mono** and 1-[(2R,3R,4S,5R,6R)-6-{[(2R,3R,4S,5R,6R)-6-{[(10-oxo-10-{[(3E,7E,11E,15E)-3,7,12,16,20-penta-methylhenicosa-3,7,11,15,19-pentaen-1-yl]oxy}-decanoyl)oxy]methyl}-3,4,5-tris[(trimethylsilyl)oxy]oxan-2-yl]oxy}-3,4,5-tris[(trimethylsilyl)oxy]-oxan-2-yl]methyl10-(3E,7E,11E,15E)-3,7,12,16,20-penta-methylhenicosa-3,7,11,15,19-pentaen-1-yl decanedioate/**16-bis**. EDC·HCl (223 mg, 1.161 mmol) and DMAP (5 mg, 0.039 mmol) were added under stirring at room temperature (RT) to a solution of hexaTMS-protected trehalose **5** [40] (300 mg, 0.387 mmol) in anhydrous toluene (8.3 mL). After 30 min, carboxylated squalene-linker adduct **6** [32] (221 mg, 0.387 mmol) was added, and the reaction mixture was stirred at 50 °C overnight. Reaction monitoring (TLC, eluant: 9:1 *n*-hexane/AcOEt) confirmed the disappearance of hexaTMS-protected trehalose **5**. The solvent was then removed under reduced pressure, and the crude oil was purified by flash chromatography (silicagel, eluant: 9:1 *n*-hexane/AcOEt) to obtain pure **15-mono (**232.9 mg, 0.175 mmol, 45% yield) and pure **16-bis** (101.2 mg, 0.054 mmol, 14% yield). 

Analytical characterization. **15-mono**: *^1^H-NMR* (CDCl_3_, 400 MHz): δ(ppm) = 5.21–5.05 (m, 5H), 4.94 (t, *J* = 2.8 Hz, 2H), 4.32 (dd, *J* = 11.8, 2.1 Hz, 1H), 4.12–3.98 (m, 4H), 3.98–3.82 (m, 3H), 3.71 (dd, *J* = 6.4, 3.4 Hz, 2H), 3.54–3.41 (m, 4H), 2.41–2.26 (m, 4H), 2.14–1.96 (m, 16H), 1.79–1.67 (m, 5H), 1.62–1.58 (m, 21H), 1.37–1.29 (m, 8H), 0.22–0.11 (m, 54H). *^13^C–NMR* (CDCl_3_, 400 MHz): δ(ppm) = 173.33, 173.18, 134.88, 134.76, 134.07, 132.4, 132.00, 129.28, 129.17, 124.84, 124.57, 124.37, 93.83, 93.67, 73.77, 73.66, 73.03, 72.05, 71.94, 70.59, 70.10, 68.28, 63.61, 63.24, 61.27, 39.73 (2C), 35.72, 33.98, 33.87, 29.45, 29.04 (3C), 28.99, 28.86, 28.18, 26.68, 26.43 (2C), 25.95, 25.01, 24.91, 22.38 (2C), 17.95, 16.28, 16.22, 16.05, (1.35, 1.18, 0.36 = 18C). *HR-ESI-MS*: MW 1349.7973 calcd. for C_67_H_130_O_14_Si_6_Na, MW 1349.7982 found. Optical rotation, ${\left[\alpha \right]}_{D}^{20}$: −61.9°. **16-bis**: *^1^H-NMR* ((CDCl_3_, 400 MHz): δ(ppm) = 5.16–5.10 (m, 10H), 4.94 (d, *J* = 3.0 Hz, 2H), 4.31–4.28 (m, 2H), 4.10–3.99 (d, *J* = 38.8 Hz, 8H), 3.95–3.90 (m, 2H), 3.52–3.44 (m, 4H), 2.38–2.28 (m, 8H), 2.13–1.98 (m, 36H), 1.77–1.68 (m, 10H), 1.66–1.59 (m, 38H), 1.36–1.28 (m, 16H), 0.21–0.11 (m, 54H). *^13^C-NMR* (CDCl_3_, 400 MHz): δ(ppm) = 173.78 (2C), 173.62 (2C), 135.05 (2C), 134.90 (2C), 134.82 (2C), 133.65 (2C), 131.15 (2C), 125.07 (2C), 124.40 (4C), 124.29 (4C), 94.40 (2C), 73.48 (2C), 72.67 (2C), 71.94 (2C), 70.74 (2C), 63.94 (2C), 63.30 (2C), 39.73 (4C), 35.80 (2C), 34.33 (2C), 34.09 (2C), 29.68 (2C), 29.11 (6C), 28.77 (2C), 28.25 (4C), 26.91 (2C), 26.76 (2C), 26.65 (4C), 25.67 (4C), 24.96 (2C), 24.74 (2C), 17.66 (2C), 16.02 (4C), 15.85 (2C), (1.05, 0.87, 0.44, 0.17 = 18C). *HR-ESI-MS*: MW 1902.2516 calcd. for C_104_H_190_O_17_Si_6_Na, MW 1902.2524 found. Optical rotation, ${\left[\alpha \right]}_{D}^{20}$: −43.9°.

1-(3E,7E,11E,15E)-3,7,12,16,20-pentamethylhenicosa-3,7,11,15,19-pentaen-1-yl 10-[(2R,3S,4S,5R,6R)-3,4,5-trihydroxy-6-{[(2R,3R,4S,5S,6R)-3,4,5-trihydroxy-6-(hydroxymethyl)oxan-2-yl]oxy}oxan-2-yl]-methyl decanedioate/**1-Sq-mono**. Acetic acid (0.1 mL, 1.73 mmol) was added under stirring at RT to a solution of **15-mono** (230.0 mg, 0.173 mmol) in MeOH (3 mL), and the reaction mixture was stirred at 40 °C overnight. Reaction monitoring (TLC, eluant: 9:1 *n*-hexane/AcOEt) confirmed the disappearance of **15-mono**. The solvent was then removed under reduced pressure, and the crude solid was purified by flash chromatography (silicagel, eluant: 85:15 CH_2_Cl_2_/MeOH) to obtain pure target **1-Sq-mono** (153.2 mg, 0.171 mmol, quantitative yield). 

Analytical characterization. *^1^H-NMR* (DMSO-d6, 400 MHz): δ(ppm) = 5.00–4.91 (m, 6H), 4.75 (d, *J* = 3.7 Hz, 2H), 4.72 (d, *J* = 3.6 Hz, 1H), 4.64 (t, *J* = 4.7 Hz, 2H), 4.55 (dd, *J* = 6.2, 1.8 Hz, 2H), 4.22 (t, *J* = 6.0 Hz, 1H), 4.13–4.10 (m, 1H), 3.92 (dd, *J* = 11.8, 5.4 Hz, 1H), 3.83 (t, *J* = 6.6 Hz, 2H), 3.80–3.76 (m, 1H), 3.55–3.51 (m, 1H), 3.46–3.40 (m, 3H), 3.38–3.32 (m, 1H), 3.16–3.10 (m, 2H), 3.04–2.98 (m, 2H), 2.1–2.12 (m, 4H), 1.95–1.79 (m, 18H), 1.56–1.49 (m, 5H), 1.44 (bs, 15H), 1.41–1.37 (m, 4H), 1.13 (bs, 8H). *^13^C-NMR* (DMSO-d6, 100 MHz): δ(ppm): δ 173.28, 173.22, 134.83, 134.77, 134.73, 133.97, 131.04, 124.82, 124.59, 124.52, 124.46, 124.35, 93.81, 93.71, 73.30, 73.26, 73.04, 72.05, 71.94, 70.59 (2C), 70.10, 63.63, 63.56, 61.27, 40.62, 39.64, 39.59, 39.37, 35.71, 33.98 (2C), 31.59, 30.28, 29.46, 29.06, 29.03, 28.91, 28.16, 26.82, 26.67, 26.42 (2C), 25.90, 24.96, 24.89, 17.94, 16.21, 16.06. *HR-ESI-MS*: MW 917.5602 calcd. for C_49_H_82_O_14_Na, MW 917.5623 found. Optical rotation, ${\left[\alpha \right]}_{D}^{20}$: −59.4°.

1-(3E,7E,11E,15E)-3,7,12,16,20-pentamethylhenicosa-3,7,11,15,19-pentaen-1-yl 10-[(2R,3S,4S,5R,6R)-3,4,5-trihydroxy-6-{[(2R,3R,4S,5S,6R)-3,4,5-trihydroxy-6-{[(10-oxo-10-{[(3E,7E,11E,15E)-3,7,12,16,20-pentamethylhenicosa-3,7,11,15,19-pentaen-1-yl]oxy}decanoyl)oxy]methyl}oxan-2-yl]oxy}oxan-2-yl]-methyl decanedioate/**2-Sq-bis**. Acetic acid (30 μL, 0.53 mmol) was added under stirring at RT to a solution of **16-bis** (100.0 mg, 0.053 mmol) in MeOH (1 mL), and the reaction mixture was stirred at 40 °C overnight. Reaction monitoring (TLC, eluant: 9:1 *n*-hexane/AcOEt) confirmed the disappearance of **16-bis**. The solvent was then removed under reduced pressure, and the crude solid was purified by flash chromatography (silicagel, eluant: 85:15 CH_2_Cl_2_/MeOH) to obtain pure target **2-Sq-bis** (69.1 mg, 0.047 mmol, 90% yield). Analytical characterization. *^1^H-NMR* (DMSO-d6, 400 MHz): δ(ppm) = 5.12–5.04 (m, 12H), 4.88 (d, *J* = 4.9 Hz, 2H), 4.83 (d, *J* = 3.6 Hz, 2H), 4.75 (d, *J* = 6.1 Hz, 2H), 4.25–4.21 (m, 2H), 4.03 (dd, *J* = 11.7, 5.6 Hz, 2H), 3.95 (t, *J* = 6.6 Hz, 4H), 3.92–3.87 (m, 2H), 3.58–3.52 (m, 2H), 3.28–3.23 (m, 2H), 3.15–3.09 (m, 2H), 2.28–2.23 (m, 8H), 2.07–1.90 (m, 36H), 1.67–1.60 (m, 10H), 1.55–1.47 (m, 38H), 1.24 (s, 16H). *^13^C-NMR* (DMSO-d6, 100 MHz): δ(ppm): 173.90 (2C), 173.64 (2C), 134.81 (2C), 134.72 (2C), 134.64 (2C), 133.63 (2C), 131.08 (2C), 124.94 (2C), 124.57 (2C), 124.52 (2C), 124.47 (2C), 124.32 (2C), 93.75 (2C), 73.52 (2C), 72.56 (2C), 71.93 (2C), 70.71 (2C),63.99 (2C), 63.34 (2C), 39.74 (4C), 39.65 (2C), 35.86 (2C), 34.38 (2C), 34.12 (2C), 29.74 (2C), 29.17 (2C), 29.10 (4C), 28.28 (4C), 26.92 (2C), 26.81 (2C), 26.68 (2C), 26.65 (2C), 25.67 (4C), 24.98 (2C), 24.90 (2C), 17.94 (2C), 16.24 (4C), 16.06 (2C). *HR-ESI-MS*: MW 1470.0147 calcd. for C_86_H_142_O_17_Na, MW 1470.0146 found. Optical rotation, ${\left[\alpha \right]}_{D}^{20}$: −33.7°.

### 2.3. Synthesis—Betulinic Acid-trehalose Conjugate **3–Be-mono**

Methyl(1R,3aS,5aR,5bR,9S,11aR)-9-hydroxy-5a,5b,8,8,11a-pentamethyl-1-(prop-1-en-2-yl)-icosa-hydro-1H-cyclopenta[a]chrysene-3a-carboxylate/**18.** Trimethylsilyl diazomethane (2M in *n*-hexane, 0.66 mL, 1.312 mmol) was added to a solution of betulinic acid **17** (500 mg, 1.093 mmol) in dry MeOH (10 mL) and dry toluene (15 mL). The reaction was stirred overnight at RT, and reaction monitoring (TLC, eluant 7:3 *n*-hexane/AcOEt with 1% HCOOH) confirmed the disappearance of starting material **16**. The reaction mixture was diluted with diethyl ether (13 mL) and 10% AcOH (10 mL). The aqueous layer was extracted with diethyl ether (3 × 10 mL), and the collected organic phases were washed with sat. Na_2_CO_3_ (10 mL), dried with Na_2_SO_4_ and then evaporated under reduced pressure to obtain pure **18** as a white solid (486.1 mg, 1.032 mmol, 95% yield). 

Analytical characterization. *^1^H-NMR* (CDCl_3,_ 400 MHz): δ(ppm) = 4.63 (bs, 1H), 4.49 (bs, 1H), 3.56 (s, 3H), 3.07 (dd, *J* = 11.2, 5.1 Hz, 1H), 2.89 (td, *J* = 10.9, 4.4 Hz, 1H), 2.20–2.02 (m, 2H), 1.77 (dt, *J* = 10.9, 5.9 Hz, 2H), 1.58 (s, 3H), 0.86 (s, 6H), 0.81 (s, 3H), 0.71 (s, 3H), 0.65 (s, 3H). *HR-ESI-MS*: MW 493.3658 calcd. for C_31_H_50_O_3_Na, MW 493.3661 found. Optical rotation, ${\left[\alpha \right]}_{D}^{20}$: +5.1°.

10-oxo-10-[2-(trimethylsilyl)ethoxy]decanoic acid/**19**. Trimethylsilylethanol (313 mL, 2.181 mmol), EDC.HCl (559 mg, 2.909 mmol) and DMAP (89 mg, 0.727 mmol) were added under stirring at RT to a solution of sebacic acid **14** (1 g, 0.4942 mmol) in dry CH_2_Cl_2_ (25 mL) and pyridine (2.5 mL). The reaction mixture was stirred at RT overnight. The reaction mixture was then washed with 10% phosphoric acid (2 × 15 mL) and brine (20 mL). 

The organic layer was dried with Na_2_SO_4_ and evaporated under reduced pressure, and the crude oil was purified by flash chromatography (silicagel, eluant: 8:2 *n*-hexane/AcOEt with 1% HCOOH) to obtain pure **19 (**408 mg, 1.342 mmol, 27% yield). 

Analytical characterization. *^1^H-NMR* (CDCl_3_, 400 MHz): δ(ppm) = 4.21–4.12 (m, 2H), 2.38 (t, *J* = 4.5 Hz, 2H), 2.32 (t, *J* = 4.3 Hz, 2H), 1.73–1.59 (m, 4H), 1.41–1.28 (m, 8H), 1.01–0.97 (m, 2H), 0.04 (s, 9H). *^13^C-NMR*: (CDCl_3_, 100 MHz): δ(ppm) = 177.93, 173.60, 62.58, 34.32, 34.22, 29.46, 29.48, 29.30, 29.10, 25.12, 25.07, 17.05, −1.53 (3C). *HR-ESI-MS*: MW 325.1811 calcd. for C_15_H_30_O_4_SiNa, MW 325.1815 found.

(1R,3aS,5aR,5bR,9S,11aR)-3a-(methoxycarbonyl)-5a,5b,8,8,11a-pentamethyl-1-(prop-1-en-2-yl)-icosahydro-1H-cyclopenta[a]chrysen-9-yl 1-[2-(trimethylsilyl)ethyl] decanedioate/**20**. Dicyclohexylcarbodiimide (DCC, 201 mg, 0.973 mmol) and dimethylaminopyridine (DMAP, 30 mg, 0.243 mmol) were added under stirring to a solution of compound **18** (229 mg, 0.487 mmol) and compound **19** (221 mg, 0.731 mmol) in dry CH_2_Cl_2_ (5 mL) at 0 °C. The reaction was left stirring at RT overnight. Reaction monitoring (TLC, eluant: 9:1 *n*-hexane/AcOEt) confirmed the disappearance of starting material **18**. The mixture was diluted with CH_2_Cl_2_ (10 mL) and was filtered on a plug of celite. The solvent was removed under reduced pressure, and the resulting crude oil was purified by flash chromatography (silicagel, eluant: 96:4 *n*-hexane/AcOEt) to obtain pure **20 (**339 mg, 0.449 mmol, 92% yield). 

Analytical characterization. *^1^H-NMR* (CDCl_3_, 400 MHz): δ(ppm) = 4.74 (bs, 1H), 4.61 (bs, 1H), 4.48 (dd, *J* = 10.1, 6.2 Hz, 1H), 4.21–4.12 (m, 2H), 3.67 (s, 3H), 3.06–2.95 (m, 1H), 2.33–2.13 (m, 6H), 1.97–1.82 (m, 2H), 1.69 (s, 3H), 1.47–1.34 (m, 8H), 0.97 (s, 3H), 0.92 (s, 3H), 0.85 (s, 3H), 0.84 (s, 6H), 0.05 (s, 9H). *^13^C-NMR*: (CDCl_3_, 100 MHz): δ(ppm) = 176.67, 174.00, 173.64, 150.57, 109.63, 80.61, 62.37, 56.57, 55.45 (2C), 51.25, 50.46, 49.48, 47.01, 42.40, 40.70, 38.40, 38.27, 37.85, 37.12, 36.98, 34.82, 34.52, 34.27, 32.18, 30.61, 29.68, 29.10 (3C), 27.97, 25.49, 25.12, 24.95, 23.75, 20.91, 19.36, 18.19, 17.33, 16.57, 16.18, 15.96, 14.69, −1.47 (3C). *HR-ESI-MS*: MW 777.5465 calcd. for C_46_H_78_O_6_SiNa, MW 777.5469 found. Optical rotation, ${\left[\alpha \right]}_{D}^{20}$: +10.2°.

10-{[(1R,3aS,5aR,5bR,9S,11aR)-3a-(methoxycarbonyl)-5a,5b,8,8,11a-pentamethyl-1-(prop-1-en-2-yl)-icosahydro-1H-cyclopenta[a]chrysen-9-yl]oxy}-10-oxodecanoic acid/**21**. Tetrabutylammonium fluoride (TBAF, 0.61 mL, 2.11 mmol) was added under stirring to a solution of compound **19** (318 mg, 0.421 mmol) in dry THF (15 mL), and the reaction mixture was stirred at RT overnight. Reaction monitoring (TLC, eluant: 9:1 *n*-hexane/AcOEt with 1% HCOOH) confirmed the disappearance of starting material **20**. The reaction was quenched by an addition of sat. NH_4_Cl (10 mL). The aqueous phase was extracted with AcOEt (2 × 10 mL), the collected organic phases were dried with Na_2_SO_4_ and evaporated under reduced pressure to obtain pure **21** (258 mg, 0.393 mmol, 93% yield).

Analytical characterization. *^1^H-NMR* (CDCl_3_, 400 MHz): δ(ppm) = 4.72 (bs, 1H), 4.58 (bs, 1H), 4.44 (dd, *J* = 10.1, 6.2 Hz, 1H), 3.65 (s, 3H), 3.03–2.94 (m, 1H), 2.23–2.16 (m, 6H), 1.92–1.82 (m, 2H), 1.67 (s, 3H), 1.47–1.34 (m, 8H), 0.95 (s, 3H), 0.90 (s, 3H), 0.84 (s, 3H), 0.82 (s, 6H). *^13^C-NMR* (CDCl_3_, 100 MHz): δ(ppm) 180.32, 177.30, 174.33, 151.11, 110.29, 81.31, 57.20, 56.08 (2C), 51.89, 51.10, 50.12, 47.63, 43.04, 41.34, 39.04, 38.90, 38.47, 37.76, 37.60, 35.43, 34.92, 34.70, 32.81, 31.25, 30.32, 29.68 (3C), 28.61, 26.13, 25.73, 25.29, 24.38, 21.56, 20.00, 18.84, 17.21, 16.81, 16.59, 15.33. *HR-ESI-MS*: MW 677.4757 calcd. for C_41_H_66_O_6_Na, MW 677.4761 found. Optical rotation, ${\left[\alpha \right]}_{D}^{20}$: +12.9°.

(1R,3aS,5aR,5bR,9S,11aR)-3a-(methoxycarbonyl)-5a,5b,8,8,11a-pentamethyl-1-(prop-1-en-2-yl)-icosahydro-1H-cyclopenta[a]chrysen-9-yl-1-[(2R,3R,4S,5R,6R)-6-{[(2R,3R,4S,5R,6R)-6-(hydroxy met-hyl)-3,4,5-tris[(trimethylsilyl)oxy]oxan-2-yl]oxy}-3,4,5-tris[(trimethylsilyl)oxy]oxan-2-yl]methyl decanedioate/**22-mono** and (1R,3aS,5aR,5bR,9S,11aR)-3a-(methoxycarbonyl)-5a,5b,8,8,11a-pentamethyl-1-(prop-1-en-2-yl)-icosahydro-1H-cyclopenta[a]chrysen-9-yl 1-[(2R,3R,4S,5R,6R)-6-{[(2R,3R,4S,5R,6R)-6-{[(10-{[(1R,3aS,5aR,5bR,9S,11aR)-3a-(methoxycarbonyl)-5a,5b,8,8,11a-penta-methyl-1-(prop-1-en-2-yl)-icosahydro-1H-cyclopenta[a]chrysen-9-yl]oxy}-10-oxodecanoyl)oxy]-methyl}-3,4,5-tris[(trimethylsilyl)oxy]oxan-2-yl]oxy}-3,4,5-tris[(trimethylsilyl)oxy]oxan-2-yl]methyl decanedioate/**23-bis**. EDC^.^HCl (15 mg, 0.0763 mmol) and DMAP (1 mg, 0.00763 mmol) were added under stirring at RT to a solution of hexaTMS-protected trehalose **5** [40]. (59 mg, 0.0763 mmol) in anhydrous toluene (4 mL). After 30 min, compound **21** (50 mg, 0.0763 mmol) was added, and the reaction mixture was stirred at 50 °C overnight. Reaction monitoring (TLC, eluant: 7:3 *n*-hexane/AcOEt) confirmed the disappearance of starting material **21**. The solvent was then removed under reduced pressure, and the crude oil was purified by flash chromatography (silicagel, eluant: 85:15 *n*-hexane/AcOEt) to obtain pure **22-mono (**23 mg, 0.0163 mmol, 21% yield) and pure **23-bis** (24 mg, 0.0117 mmol, 15% yield). 

Analytical characterization. **22-mono**: *^1^H-NMR* (CD_3_OD, 400 MHz): δ(ppm) = 4.96 (dd, *J* = 3.0, 1.9 Hz, 2H), 4.74 (d, *J* = 2.1 Hz, 1H), 4.67–4.58 (m, 1H), 4.47 (dd, *J* = 10.6, 5.7 Hz, 1H), 4.43–4.34 (m, 1H), 4.12–4.02 (m, 2H), 3.99 (td, *J* = 9.0, 3.1 Hz, 2H), 3.87 (dt, *J* = 9.5, 3.0 Hz, 1H), 3.70–3.68 (m, 5H), 3.61–3.45 (m, 4H), 3.34–3.32 (m, 2H), 3.02 (td, *J* = 10.8, 4.7 Hz, 1H), 2.44–2.23 (m, 6H), 1.90 (tt, *J* = 11.7, 5.8 Hz, 2H), 1.78–1.70 (m, 6H), 1.36 (s, 3H), 1.04 (s, 3H), 0.97 (s, 3H), 0.92 (s, 3H), 0.89 (s, 3H), 0.88 (s, 3H), 0.25–0.14 (m, 54H). *^13^C-NMR* (CD_3_OD, 400 MHz): δ (ppm) = 176.69, 173.89, 173.68, 150.33, 108.96, 94.45, 94.19, 80.81, 73.58, 73.54, 73.31, 72.66 (2C), 72.01, 71.21, 70.72, 62.91, 60.41, 56.48, 55.45 (2C), 50.44, 49.24, 47.09, 42.17, 40.53, 38.24, 38.19, 37.52, 36.91, 36.47, 34.16, 34.06, 33.62, 31.74, 30.24, 29.42, 28.78 (2C), 28.73, 28.62, 27.22, 25.38, 24.80, 24.58, 23.38, 20.72, 18.21, 17.90, 15.75, 15.42, 15.20, 13.83, (0.19, −0.25, −0.99, −1.07 = 18C). *HR-ESI-MS*: MW 1433.8185 calcd. for C_71_H_134_O_16_Si_6_Na, MW 1433.8191 found. Optical rotation, ${\left[\alpha \right]}_{D}^{20}$: +60.3°. **23-bis**: *^1^H-NMR* (CDCl_3_, 400 MHz, detected signals): δ(ppm) = 4.94 (d, *J* = 3.0 Hz, 2H), 4.76 (d, *J* = 2.0 Hz, 2H), 4.62 (d, *J* = 3.2 Hz, 2H), 4.53–4.44 (m, 2H), 4.29 (dd, *J* = 11.8, 2.0 Hz, 2H), 4.07 (dd, *J* = 11.8, 4.3 Hz, 2H), 4.02 (ddd, *J* = 9.4, 4.2, 2.1 Hz, 2H), 3.92 (t, *J* = 9.0 Hz, 2H), 3.69 (s, 6H), 3.50 (t, *J* = 9.0 Hz, 2H), 3.46 (dd, *J* = 9.3, 3.1 Hz, 2H), 3.01 (td, *J* = 10.8, 4.2 Hz, 2H), 2.40–2.16 (m, 12H), 1.98–1.83 (m, 4H), 1.71 (s, 6H), 1.68–1.56 (m, 16H), 1.28 (s, 6H), 0.98 (s, 6H), 0.93 (s, 6H), 0.86 (s, 6H), 0.85 (s, 12H), 0.16 (m, 54H).

*^13^C-NMR* (CDCl_3_, 100 MHz): δ (ppm) = 176.66 (2C), 173.68 (2C), 173.60 (2C), 150.55 (2C), 109.62 (2C), 94.44 (2C), 80.61 (2C), 73.49 (2C), 72.67 (2C), 71.94 (2C), 70.75 (2C), 63.31 (2C), 56.57 (2C), 55.46 (2C), 51.23 (2C), 50.47 (2C), 49.49 (2C), 47.01 (2C), 42.40 (2C), 40.71 (2C), 38.40 (2C), 38.27 (2C), 37.84 (2C), 37.13 (2C), 36.97 (2C), 34.81 (2C), 34.28 (2C), 34.11 (2C), 32.18 (2C), 30.61 (2C), 29.68 (2C), 29.14 (2C), 29.11 (2C), 29.09 (2C), 27.97 (2C), 25.49 (2C), 25.12 (2C), 24.75 (2C), 23.75 (2C), 20.91 (2C), 19.35 (2C), 18.19 (2C), 16.56 (2C), 16.16 (2C), 15.96 (2C), 14.69 (2C), 14.10 (2C), (1.06, 0.88, 0.18 =18C). *HR-ESI-MS*: MW 2070.2939 calcd. for C_112_H_198_O_21_Si_6_Na, MW 2070.2949 found. Optical rotation, ${\left[\alpha \right]}_{D}^{20}$: +41.2°.

(1R,3aS,5aR,5bR,9S,11aR)-3a-(methoxycarbonyl)-5a,5b,8,8,11a-pentamethyl-1-(prop-1-en-2-yl)-icosahydro-1H-cyclopenta[a]chrysen-9-yl 1-[(2R,3S,4S,5R,6R)-3,4,5-trihydroxy-6-{[(2R,3R,4S,5S,6R)-3,4,5-trihydroxy-6-(hydroxymethyl)oxan-2-yl]oxy}oxan-2-yl]methyl decanedioate/**3-Be-mono**. Acetic acid (18 μL, 0.324 mmol) was added under stirring at RT to a solution of **22-mono** (23 mg, 0.0162 mmol) in MeOH (1 mL), and the reaction mixture was stirred at 40 °C for two days. Reaction monitoring (TLC, eluant: 98:2 CH_2_Cl_2_/MeOH) confirmed the disappearance of starting **22-mono**. The solvent was then removed under reduced pressure to obtain pure target **3-Be-mono** (13 mg, 0.0133 mmol, 82% yield). 

Analytical characterization. *^1^H-NMR* (CDCl_3_, 400 MHz): δ(ppm) = 4.98 (dd, *J* = 3.0, 1.9 Hz, 2H), 4.76 (d, *J* = 2.0 Hz, 1H), 4.65–4.59 (m, 1H), 4.52–4.42 (m, 1H), 4.45–4.37 (m, 1H), 4.11–4.02 (m, 2H), 3.97 (td, *J* = 8.9, 3.1 Hz, 2H), 3.84 (dt, *J* = 9.4, 3.1 Hz, 1H), 3.72–3.67 (m, 5H), 3.60–3.43 (m, 4H), 3.35–3.33 (m, 2H), 3.01 (td, *J* = 10.8, 4.7 Hz, 1H), 2.43–2.17 (m, 6H), 1.98–1.85 (m, 2H), 1.75–1.68 (m, 6H), 1.28 (s, 3H), 0.98 (s, 3H), 0.94 (s, 3H), 0.87 (s, 3H), 0.85 (s, 6H). *^13^C-NMR* (CDCl_3_, 100 MHz): δ(ppm) = 176.64, 173.80, 173.72, 150.50, 109.65, 94.42, 94.15, 80.74, 73.59, 73.51, 73.30, 72.66, 72.62, 72.04, 71.25, 70.68, 62.90, 60.37, 56.56, 55.44, 51.24, 50.46, 49.49, 47.00, 42.40, 40.71, 38.40, 38.26, 37.85, 37.13, 36.96, 34.82, 34.28, 34.18, 34.12, 34.09, 32.18, 30.62, 29.69, 29.35, 29.17, 28.01, 25.48, 25.14, 24.81, 23.75, 20.92, 19.38, 18.21, 16.61, 16.18, 15.96, 14.71. *HR-ESI-MS*: MW 1001.5813 calcd. for C_53_H_86_O_16_Na, MW 1001.5819 found. Optical rotation, ${\left[\alpha \right]}_{D}^{20}$: +70.6°.

Attempted synthesis of (1R,3aS,5aR,5bR,9S,11aR)-3a-(methoxycarbonyl)-5a,5b,8,8,11a-pentamethyl-1-(prop-1-en-2-yl)-icosahydro-1H-cyclopenta[a]chrysen-9-yl 1-[(2R,3S,4S,5R,6R)-6- {[(2R,3R,4S,5S,6R)-6-{[(10-{[(1R,3aS,5aR,5bR,9S,11aR)-3a-(methoxycarbonyl)-5a,5b,8,8,11a-pentamethyl-1-(prop-1-en-2-yl)-icosahydro-1H-cyclopenta[a]chrysen-9-yl]oxy}-10-oxodecanoyl)oxy]methyl}-3,4,5-trihydroxyoxan-2-yl]oxy}-3,4,5-trihydroxyoxan-2-yl]methyl decanedioate/**4-Be-bis**. Acetic acid (13 μL, 0.234 mmol) was added under stirring at RT to a solution of **23-bis** (24 mg, 0.0117 mmol) in MeOH (1 mL), and the reaction mixture was stirred at 40 °C for four days. Reaction monitoring (TLC, eluant: 98:2 CH_2_Cl_2_/MeOH) showed the formation of a series of uncharacterizable degradation products.

### 2.4. NA Assembly and Characterization

**Mono****-****Sq-NA1**. In accordance with standard solvent evaporation protocols [41] the squalene-trehalose conjugate **1-mono** (4.0 mg) was first dissolved in THF (1 mL) in a vial while stirring at RT. The resulting solution was added dropwise to a round bottom flask containing MilliQ grade distilled water (2 mL) under magnetic stirring (500 rpm). The resulting suspension was stirred for 5 min, then THF was thoroughly evaporated under reduced pressure, obtaining pure **mono****-Sq-NA1** as an opalescent suspension (2 mL, 2 mg/mL).

**Bis-Sq-NA2**. In accordance with standard solvent evaporation protocols [41] the squalene-trehalose conjugate **2-bis** (4.0 mg) was first dissolved in THF (1 mL) in a vial while stirring at RT. The resulting solution was added dropwise to a round bottom flask containing MilliQ grade distilled water (2 mL) under magnetic stirring (500 rpm). The resulting suspension was stirred for 5 min, then THF was thoroughly evaporated under reduced pressure, obtaining pure **bis-Sq-NA2** as opalescent suspension (2 mL, 2 mg/mL).

**Mono-Be-NA3**. In accordance with standard solvent evaporation protocols [41] the betulinic acid-trehalose conjugate **3-mono** (4.0 mg) was first dissolved in THF (1 mL) in a vial while stirring at RT. The resulting solution was added dropwise to a round bottom flask containing MilliQ grade distilled water (2 mL) under magnetic stirring (500 rpm). The resulting suspension was stirred for 5 min, then THF was thoroughly evaporated under reduced pressure, obtaining pure **mono-Be-NA3** as opalescent suspension (2 mL, 2 mg/mL).

**NA Characterization**. NAs were characterized by dynamic light scattering (DLS), using a 90 Plus Particle Size Analyzer from Brookhaven Instrument Corporation (Holtsville, NY, USA) operating at 15 mW of a solid-state laser (λ = 661 nm), using a 90-degree scattering angle. The ζ-potential was determined at 25 °C using a 90 Plus Particle Size Analyzer from Brookhaven Instrument Corporation (Holtsville, NY, USA) equipped with an AQ-809 electrode, operating at an applied voltage of 120 V. Each sample was diluted to a concentration of 0.2 mg/mL and sonicated for 3 min before each experiment. Ten independent measurements of 60 s duration were performed for each sample. Hydrodynamic diameters were calculated using Mie theory, considering the absolute viscosity and refractive index values of the medium to be 0.890 cP and 1.33, respectively. The same aqueous samples at a concentration of 0.2 mg/mL were used for ζ-potential measurement, without any change for the ionic strength (no addition of KCl). The ζ-potential was calculated from the electrophoretic mobility of nanoparticles, by using the Smoluchowski theory [42].

### 2.5. Biology

Cell cultures. HeLa cells (ATCC: CCL-2) were cultured in DMEM with 10% FBS, 1% penicillin/streptomicin and 1% glutamine in a humidified atmosphere of 5% CO_2_ at 37 °C (all reagents from Euroclone). Cultures were treated with lipid-trehalose conjugates or NAs for 2 or 48 h at 37 °C at the concentration indicated in the text.

Cytotoxicity assay. We performed the 3-(4,5-dimethylthiazol-2-yl)-2,5-diphenyltetrazolium bromide (MTT) assay to measure culture vitality. HeLa cells were cultured in a 96-well plate at a concentration of 5 × 10^3^ cell/cm^2^ and incubated at 37 °C for 24 h. MTT was added in cell medium at a final concentration of 0.25 mg/mL. Incubation lasted 30 min at 37 °C. Then, the medium was removed and formazan precipitates were collected in 200 μL of DMSO. The absorbance measured at 570 nm using a spectrophotometer reflects cell viability. Cell viability was expressed as fold over control condition set at 100%.

Autophagy assay. We assessed autophagy by monitoring LC3 conversion by western-blotting as previously described [43]. Briefly, upon a wash in PBS, cells were solubilized in RIPA buffer (150 mM NaCl, 50 mM HEPES, 0.5% NP40, 1% sodium-deoxycholate). After 1 h under mild agitation, the lysate was clarified by centrifugation for 20 min at 16,000 g. All experimental procedures were performed at 4 °C. Protein concentrations were evaluated via Bradford assay (Bio-Rad, Segrate, Italy). For Western blotting experiments, an equal amount of proteins was diluted with 0.25% 5X Laemmli buffer, separated onto 10% SDS-PAGE gels and transferred onto nitrocellulose membrane (Sigma-Aldrich Italy, Milan, Italy) at 80 V for 120 min at 4 °C. 

Primary antibodies (source in parentheses) included: Mouse anti-LC3, 1:500 (Enzo Life Sciences AG, Lausen, Switzerland), and mouse anti β-actin 1:1000 (Sigma Aldrich Italy, Milan, Italy) which were applied overnight in blocking buffer (20 mM Tris, pH 7.4, 150 mM NaCl, 0.1% Tween 20, and 5% nonfat dry milk). Proteins were detected using the ECL prime detection system (GE Healthcare). Images were acquired with the imaging ChemiDoc Touch system (Bio Rad Laboratory Italy, Segrate, Italy), and the optical density of the specific bands was measured with ImageLab software (Bio Rad).

### 2.6. Statistical Analysis

All data are reported as mean ± standard error of the mean (SEM). The entire data-set was logged into GraphPad Prism and analyzed via unpaired Student’s T-test (two classes) or ANOVA followed by Tukey’s posthoc test (more than two classes). Number of experiments (n) and level of significance (p) are indicated throughout the text.

## 3. Results

### 3.1. Synthesis of Target **1-mono** and **2-bis** Squalene-Trehalose Conjugates

In order to obtain either a mono-(target compound **1-Sq-mono**, 1:1 squalene-trehalose conjugate) and a bis-squalenylated trehalose construct (target compound **2-Sq-bis**, 2:1 squalene-trehalose conjugate), we focused our attention onto the hexaTMS-protected trehalose derivative **5** [38] and the carboxylated squalene-linker adduct **6 [31]** (Figure 2).

The synthesis of key intermediates **5** [38] and **6** [31] is reported respectively in Scheme 1 and Scheme 2.

HexaTMS-protected trehalose **5** was obtained through per-silylation of commercially-available trehalose **7** with TMS-Cl (per-silylated **8**, step a), followed by a selective deprotection of primary, more easily accessible, hydroxyls (step b, Scheme 1).

Commercial squalene 9 was sequentially submitted to halohydration (bromohydrine 10, step a), base-promoted elimination (epoxide 11, step b), oxidative cleavage (aldehyde 12), reduction (alcohol 13) and mono-esterification with diacid 14, to provide target carboxylated squalene adduct 6 (step e, Scheme 2).

Finally, key intermediates **5** and **6** were coupled in equimolar amounts in an esterification protocol, obtaining a ≈ 3:1 mixture of **15-mono** and **16-bis** hexaTMS-protected compounds (step a, Scheme 3).

After chromatographic separation, both hexaTMS protected compounds **15-mono** and **16-bis** were submitted to acidic deprotection, yielding respectively **1-Sq-mono** and **2-Sq-bis** targets, respectively in **45%** and **13%** overall yields from **5** and **6** (step b, Scheme 3).

### 3.2. Synthesis of Target **3-Be-mono** Betulinic Acid-Trehalose Conjugate, Attempted Synthesis of Target **4-Be bis** Betulinic Acid-Trehalose Conjugate

In order to obtain the target **3-Be-mono** betulinic acid-trehalose conjugate we adopted a similar strategy, focusing our attention onto the same hexaTMS-protected trehalose derivative **5** [38] and the carboxylated betulinic-linker adduct **21** (Figure 3).

The synthesis of key intermediate **21** is reported in Scheme 4.

Betulinic acid **17** was first esterified (methyl ester **18**, step a), then coupled with mono-protected diacid **19** (prepared by a controlled esterification of sebacic acid **14**, step b) to provide silyl-protected construct **20** (step c). Carboxylic acid deprotection finally provided target carboxylated betulinic-linker adduct **21** (step d, Scheme 4).

Finally, key intermediates **5** and **21** were coupled in equimolar amounts in an esterification protocol, obtaining a ≈ 1.5:1 mixture of **22-mono** and **23-bis** hexaTMS-protected compounds (step a, Scheme 5).

After chromatographic separation, hexaTMS protected compound **22-mono** was submitted to acidic deprotection, yielding target **3-Be-mono** betulinic acid–trehalose conjugate, in 21% overall yield from **5** and **21** (step b, Scheme 5). The same reaction, targeting **4-Be-bis** from **23-bis**, leads to uncharacterizable degradation products.

### 3.3. NA Assembly and Structural Characterization

Both **1-Sq-mono** and **2-Sq-bis** squalene-trehalose conjugates, and **3-Be-mono** betulinic acid-trehalose conjugate were assembled into their corresponding NAs (**mono-Sq-NA1**, left, **bis-Sq-NA2**, middle, and **mono-Be-NA3**, right, Scheme 6) following a standard experimental protocol [38].

Self-assembled **mono-Sq-NA1**, **bis-Sq-NA2** and **mono-Be-NA3** were characterized in terms of hydrodynamic diameter and ζ-potential, as shown in Table 1.

The size of the hydrodynamic diameters shows an increase from **mono-Sq-NA1** to **bis-Sq-NA2**, while the self-assembly of **mono-Be-NA3** results in much larger NAs with a mean HD centered at about 460 nm. However, the polydispersity index confirms the mono-dispersion of the colloidal solution of each NA. Moreover, the self-assembled NAs show good colloidal stability as confirmed by their ζ-potential value (<−20.0 mV), and by the stability of their hydrodynamic diameter (HD) which is not affected even after 10 days’ storage in aqueous solution.

Furthermore, the TEM images and UV spectra of **mono-Sq-NA1** (respectively Appendix A), of **bis-Sq-NA2** (Appendix A) and of **mono-Be-NA3** (Appendix A) are provided in the Appendix A.

### 3.4. Biological Profiling

Finally, the set of three NAs was submitted to biological profiling in HeLa cells for cytotoxicity (safety determination) and autophagy induction (activity determination). Namely, we treated HeLa cultures for 2 and 48 h at 37 °C with either the three NAs (either estimated, adjusted 20 μM concentrations of trehalose in water for 2 h, or estimated, adjusted 40 μM concentrations for 48 h), their non-assembled squalene-trehalose precursors **1-Sq-mono** and **2-Sq-bis** (either 20 μM in DMSO for 2 h, or 40 μM for 48 h) and betulinic acid-trehalose precursor **3-Be-mono** (either 20 μM in EtOH for 2 h, or 40 μM for 48 h), or each individual component (100 mM trehalose in water, either 20 μM squalene in DMSO and 20 μM betulinic acid in EtOH for 2 h, or 40 μM with both for 48 h), and relative vehicle (DMSO or EtOH). Assays were carried out at 48 h to ensure the release of free trehalose from NAs.

At first, we determined the in vitro safety profile of each sample via the MTT cytotoxicity assay (Figure 4).

The tested samples do not elicit an overt toxicity upon 2 hours of treatment. Instead, while the set of three NAs confirmed lack of cytotoxicity at 48 h, both the **2-Sq-bis** construct (≈65% viable cells at 48 h) and free betulinic acid (≈25% viable cells at 48 h) show significant cytotoxicity (*p* < 0.001 versus not treated, *n* = 4).

Next, we assessed if NAs, non-assembled precursors and individual components, could induce autophagy by western-blotting. In accordance with cytotox results, we tested NAs at both timelines (2 h/Figure 5, and 48 h/Figure 6), while non-assembled precursors and individual components were tested only at 2 h. Autophagy can be monitored by tracking the mobility shift from LC3I to LC3II (Figure 5 and Figure 6, right), that is a bona fide reporter of the induction of autophagy; and by the amount of LC3II, that correlates with the formation of autophagosomes (Figure 5 and Figure 6, left). α-Tubulin was used as an internal control in the assays.

At 2 h, we only observed a moderate effect by the **2-Sq-bis** construct that did not reach statistical significance (Figure 5, left and right). At 48 h, surprisingly, the three NAs did not show any effect on autophagy induction/progress (Figure 6).

We investigated the fate of our NAs in the biological medium, to rationalize their lack of biological effects. Thus, HeLa cell lysates were treated following a published procedure [44], obtaining two protein-free aqueous (≈4:3 MeOH:water) and organic (≈3:1 MeOH:chloroform) layers. Their LC-MS analysis could neither detect trehalose as such at the expected μM concentration, nor the most likely lipid-trehalose intermediates (see Appendix A, **mono-Sq-NA1**; and Appendix A
**mono-Be-NA3**, Appendix A). We could not rule out the presence of trehalose at lower, nM concentrations that would not elicit an autophagy-inducing effect in cells, due to its detection LC-MS limits; and we suggest that highly lipophilic lipid-trehalose intermediates **1-Sq-mono** and **3-Be-mono** remain trapped within the protein pellet.

## 4. Discussion

Accumulating evidence indicates that induction of autophagy can be clinically relevant in the context of neurodegenerative disorders characterized by protein aggregation [39]. Pre-clinically, trehalose alleviates protein aggregation and cellular toxicity in pathological deposition of amyloid/Alzheimer [22] and alpha synuclein/Parkinson [20]. It may act by binding to extra-cellular GLUT transporters and by inducing AMPK-dependent autophagy [19], and/or by cytosolic activation of the TFEP pathway [45]. Trehalose is highly hydrophilic, preventing its passive cell permeation [46]. Moreover, trehalases hydrolyze it to glucose at the GI barrier [23]. Thus, trehalose PK in humans is challenging. By conjugating trehalose with squalene and betulinic acid, and self-assembling three constructs into NAs, we generated entities with a putatively higher permeability profile, hopefully leading to higher effects on autophagy at lower dosages.

The 2:1 **2-Sq-bis** construct showed per se an indication of higher potency than trehalose on autophagy induction at much lower, ≈40 μM, concentrations. We hypothesize that its higher lipophilicity, compared to its 1:1 **1-Sq-mono** and **3-Be-mono** counterparts, may yield better cell permeability after 2 hours of incubation, with a significant effect on autophagy induction.

The three NAs did not show any cytotoxicity, supporting their testing in the autophagy induction assay. The three **mono-Sq-NA1**, **bis-Sq-NA2** and **mono-Be-NA3** NAs did not show any effect upon autophagy induction; their inactivity could be justified by the absence of μM-free trehalose in cell lysates (LC-MS determination), possibly due to limited degradation of NAs in 48 h.

Our next efforts will include the design and execution of modified assays to measure autophagy induction at longer times and/or more NA degradation-prone conditions, and the synthesis of modified self-assembly trehalose-nanolipid conjugates to fine-tune their properties.

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
