# Peer review of "Nanolipid-Trehalose Conjugates and Nano-Assemblies as Putative Autophagy Inducers"

_pharmaceutics, 2019, doi:10.3390/pharmaceutics11080422_

Round 1
Reviewer 1 Report
The synthesis of trehalose-based lipids, and their nano-assemblies and examination for autophage inducers are interestig. The manuscript provides extensive contents for synthesis and need to further carry out and provide data regarding the nano-assemblies and corresponding discussion about relationship between the nanoassembly structures and the authophagy inducing properties. UV absorption spectra, SEM, AFM or TEM images should be fine additions to this manuscript. The manuscript seems acceptable after responding to these points.
Author Response
The synthesis of trehalose-based lipids, and their nano-assemblies and examination for autophagy inducers are interesting. The manuscript provides extensive contents for synthesis and need to further carry out and provide data regarding the nano-assemblies and corresponding discussion about relationship between the nanoassembly structures and the authophagy inducing properties.
We have added experimental data about the nano-assemblies, and we have expanded the discussion about their biological properties (or lack thereof); we’re still planning to introduce analytical data / experiments (characterization of nanoassembly-treated cell lysates, currently ongoing), so to justify the biological results.
UV absorption spectra, SEM, AFM or TEM images should be fine additions to this manuscript.
Done, TEM.
Reviewer 2 Report
This study describes the synthesis tetralose-conjugates with squalene and betulinic acid, which were used to produce nano-assemblies (NAs). The synthesis is well executed and the desired compounds and resulting NAs are well characterized.
The NAs essentially failed to show any effect on autophagy.
The authors fail to explain the envisaged fate of the NAs. Just improve the PK, but how and were to they expect the tetralose to be released.
In their discussion the authors overinterpret the results of precursor 2-Sq-bis, which they earlier called statistically not significant!
Furthermore, they write that assays were carried out to ensure the release of free tetralose from the NAs but I cannot find any evidence for that (what assays, results?).
They use the term bioavailability, which is not appropriate for the performed in vitro assays.
Overall, I find the biological results and the discussion in the present version of the manuscript poor and the authors should address these issues before acceptance can be considered.
Author Response
This study describes the synthesis tetralose-conjugates with squalene and betulinic acid, which were used to produce nano-assemblies (NAs). The synthesis is well executed and the desired compounds and resulting NAs are well characterized.
The NAs essentially failed to show any effect on autophagy.
Unfortunately, this is true. The ongoing / to be added analytical experiments on nanoassembly-treated cell lysates should allow a better interpretation of the results.
The authors fail to explain the envisaged fate of the NAs. Just improve the PK, but how and were to they expect the tetralose to be released.
We were expecting the nanoassemblies to be able to cross cell membranes as such, and to be degraded intra-cellularly. The ongoing analytical experiments should allow us to determine, at least partially, their fate in the biological profiling experiments.
In their discussion the authors overinterpret the results of precursor 2-Sq-bis, which they earlier called statistically not significant!
Although we believe this result / nano assembly to show a significant trend vs. activity, we followed the Reviewer’s suggestion by down-playing such result in the discussion.
Furthermore, they write that assays were carried out to ensure the release of free tetralose from the NAs but I cannot find any evidence for that (what assays, results?).
Such bioanalytical assays are currently ongoing, as mentioned above; their results will be used to answer this, and other comments by the Reviewers.
They use the term bioavailability, which is not appropriate for the performed in vitro assays.
Thanks for the suggestion, amended as cell permeability.
Overall, I find the biological results and the discussion in the present version of the manuscript poor and the authors should address these issues before acceptance can be considered.
While we concur that the biological results were not as expected in terms of efficacy / potency on autophagy, the experiments were well conceived, and technically properly executed. The bioanalytical ongoing experiments should allow the rationalization of observed effects; thus, we believe that this concern of the Reviewer should be now addressed.
Reviewer 3 Report
please find attachment.

Author Response
In the manuscript “Nanolipid-Trehalose Conjugates and Nano-Assemblies as Putative Autophagy Inducers”, authors have overcome the pharmacokinetic challenges of trehalose to effectively use it as autophagy inducer. I have following observations and suggestions:
A brief introduction and importance of autophagy should be given
The intro on autophagy was slightly enlarged, as to the Reviewer’s suggestion
‘stirring at rt’, where the abbreviation ‘rt’ is defined?
Rt was defined in the text
Line 412: ‘mean HD’, where is HD defined?
HD is defined in the Table heading (hydrodynamic diameter)
Discussion part is very weak, the results are not comprehensively explained and analyzed. Discussion is the heart of the manuscript and should be written very convincingly, justifying the experiments and all the results. Authors have not even mentioned about the cytotoxicity in the discussion part.
We briefly mentioned the lack of toxicity for all nano-assemblies, and we plan to expand the discussion once the current analytical experiments on nanoassembly-treated cell lysates will be completed. We believe that the result should be acceptable for publication.
In the last lines of the discussion, authors have mentioned about their future plans. This should be separately written in a paragraph ‘future prospects’, and in addition to those points, authors should also mention about other important aspects such as determining the mechanisms of action and in-vivo activity.
We have referred to previous comments on currently ongoing analytical experiments, and we plan to clarify them in the text; thus, we believe that a “Future Prospects” Heading is not justified.
At very few places, grammatical corrections are needed, such as: Abstract: ‘in a 1:2 and a 1:1 stoichiometry’, ‘a’ should be removed. Line 41: ‘and to trehalase enzymes’, either ‘to’ should be removed or ‘due to’ should be written. Line 42: ‘preventing its oral administration’, ‘administration’ should be replaced with ‘absorption’ Line 294: diluted at a concentration’ should be ‘diluted to a concentration’
All suggestions / amendments were taken care of in the text.
Reviewer 4 Report
Colombo et al present a study on the synthesis and biological action of trehalose conjugates and nanoasseblies. Although the results of the biological studies are so far not understood and need further investigation, the results are well-presented, easy to follow and support the conclusions.
There are, however, some points, which need further attention:
- there is no motivation for the use of squalene and betulinic acid given in the introduction, besides the reference to the literature. One sentence can help the reader to get the context
- what was the sample preparation for zeta-potential measurements? Has KCl solution been added? What was the ionic strength of the dispersion?
- SEM images may help to elucidate the particle size distribution and allow the authors to give a statement, whether particle fractions of different sizes are present - which would affect cellular uptake behavior.
- what is the overall yield of the process?
- scheme 4: in step b no trimethylsilylethanol is mentioned in the figure caption
- regarding the visualization of the nano assemblies: the trehalose is clearly the most hydrophilic component of the particles, thus it will be located at the interface/surface of the particles. This should be considered for the sketches.
- how was the concentration of the trehalose adjusted. Have the authors used identical volumes of differently concentrated dispersions or adjusted volumes?
- is it possible to formulate squalen/betulinic acid nano assemblies as controls and use them instead of DMSO solutions?
- the authors use "biodegradable" linkers - ester groups, which are prone to hydrolysis. As the authors find low biological action, the absence of hydrolysis may be one of the reasons. Have the authors studied the degradation of the linkers in biological medium?
Author Response
Colombo et al present a study on the synthesis and biological action of trehalose conjugates and nanoassemblies. Although the results of the biological studies are so far not understood and need further investigation, the results are well-presented, easy to follow and support the conclusions.
There are, however, some points, which need further attention:
- there is no motivation for the use of squalene and betulinic acid given in the introduction, besides the reference to the literature. One sentence can help the reader to get the context
We’ve expanded the introduction regarding betulinic acid and squalene, as for the Reviewer’s suggestion
- what was the sample preparation for zeta-potential measurements? Has KCl solution been added? What was the ionic strength of the dispersion?
KCl solution was not added, and ionic strength was not changed. These questions were addressed in the current Manuscript draft
- SEM images may help to elucidate the particle size distribution and allow the authors to give a statement, whether particle fractions of different sizes are present - which would affect cellular uptake behavior.
We’ve acquired TEM images to enlarge the structural characterization of our nanoassemblies.
- what is the overall yield of the process?
- scheme 4: in step b no trimethylsilylethanol is mentioned in the figure caption
The correction was executed in the current draft.
- regarding the visualization of the nano assemblies: the trehalose is clearly the most hydrophilic component of the particles, thus it will be located at the interface/surface of the particles. This should be considered for the sketches.
The sketched NAs were graphically amended (Scheme 6) to better reproduce the expected nanoassembly poses
- how was the concentration of the trehalose adjusted. Have the authors used identical volumes of differently concentrated dispersions or adjusted volumes?
- is it possible to formulate squalen/betulinic acid nano assemblies as controls and use them instead of DMSO solutions?
It should be possible, but we deemed such action as less prioritary than others, as the trehalose nanoassemblies resulted to be both non toxic and inactive on autophagy
- the authors use "biodegradable" linkers - ester groups, which are prone to hydrolysis. As the authors find low biological action, the absence of hydrolysis may be one of the reasons. Have the authors studied the degradation of the linkers in biological medium?
The ongoing bioanalytical studies on nanoassembly-treated cell lysates should at least partially clarify these questions.
Round 2
Reviewer 1 Report
The manuscript has been substantially modified and is now acceptable.
Reviewer 2 Report
The authors have addressed most of the concerns raised in the first review round. Although they move many of the open questions to future prospects, I recommend acceptance of the revised manuscript for publication.
Reviewer 3 Report
Acceptable.
Reviewer 4 Report
with the changes and amendments, the manuscript is suitable for publication